# Thermodynamics of Reinforcement Learning Curricula

## Abstract

Connections between statistical mechanics and machine learning have repeatedly proven fruitful, providing insight into optimization, generalization, and representation learning. In this work, we follow this tradition by leveraging results from non-equilibrium thermodynamics to formalize curriculum learning in reinforcement learning (RL). In particular, we propose a geometric framework for RL by interpreting reward parameters as coordinates on a task manifold. We show that, by minimizing the excess thermodynamic work, optimal curricula correspond to geodesics in this task space. As an application of this framework, we provide an algorithm, "MEW" (Minimum Excess Work), to derive a principled schedule for temperature annealing in maximum-entropy RL.

## 1 Introduction

Modern reinforcement learning (RL) systems are rarely trained on a single, static task. Instead, agents are routinely exposed to sequences of related tasks through curriculum learning, temperature annealing, reward shaping, and other non-stationary objectives. Despite their ubiquity, the principles governing *how* tasks should be varied remain poorly understood. A simple and practical method is to interpolate task (i.e., reward function) parameters linearly in time. This choice implicitly assumes that the task space is flat and isotropic. In this work, we hypothesize that this assumption is false, and aim to show the existence of a nontrivial geometry induced by an agent and its learning dynamics.

In particular, we adopt a statistical mechanics-based approach to studying the space of parameterized reward functions, revealing a natural metric that quantifies the difficulty or "friction" associated with adapting to a new task. More specifically, we employ a *friction tensor*, which quantifies the cost of controlling a system in non-equilibrium statistical mechanics (NESM), such that optimal parameter protocols correspond to geodesics in the geometry induced by the friction tensor. By mapping RL onto this framework, we obtain a principled hypothesis for curriculum optimality that remains experimentally tractable: **optimal reward parameter schedules minimize the path-dependent excess cost from the friction tensor and follow geodesics in the induced task space**.

This geometric picture has the potential to unify several phenomena in RL, such as potential-based reward shaping, simulated annealing, and feature collapse. In this work, we focus on linear reward function parameterizations, and derive a closed-form expression for one-dimensional task schedules, yielding a new approach for entropy temperature annealing readily applicable to deep RL.

## 2 Background

RL-analogous ideas of "curricula" arise in the control of non-equilibrium physical systems. In the framework of statistical mechanics, system dynamics depend on externally controlled parameters (e.g. temperature, coupling strengths, field strengths, trap positions, etc.) that are varied over time. When these parameters are changed (infinitesimally) slowly, the system remains near equilibrium and the required external work for this change depends only on the endpoints. When parameters change at a finite rate, however, the system remains out of equilibrium and incurs additional, path-dependent dissipation, quantified as "excess work" Jarzynski (2008).

A central result from linear response theory shows that this excess work admits a quadratic approximation in the parameter velocities Sivak & Crooks (2012). This framework has been successfully

applied to a range of classical and quantum control problems. In this work, we show that task interpolation in RL admits an analogous geometric structure: varying reward parameters induces transient sub-optimality and learning inefficiency, and the leading-order cost of this adaptation can be characterized by a metric on task space defined by long-time policy-induced correlations.

Such connections between statistical mechanics and machine learning have historically proven useful, providing insight into optimization, generalization, and representation learning (Pennington & Worah, 2017; Yaida, 2019; Bahri et al., 2020; Barr et al., 2020; Huang, 2021; Das et al., 2021; Roberts et al., 2022; Gillman et al., 2024; Bahri et al., 2024). Our contribution follows this tradition by leveraging non-equilibrium thermodynamics to formalize curriculum learning and task interpolation in RL. We note that this framework may also find applicability in other facets of machine learning based on the universal importance of distribution shift and parameter control.

The bridge between reinforcement learning and statistical mechanics is most transparent in the maximum entropy (MaxEnt) formulation of RL, where policies are optimized to maximize expected reward and entropy, as shown below in Equation 1. We consider the standard Markov Decision Process (MDP) formulation with state space $\mathcal{S}$, action space $\mathcal{A}$, transition kernel $P(s'|s, a)$, and reward function $r_\lambda(s, a)$ parameterized by $\lambda \in \mathbb{R}^L$. We assume that stationary optimal policies $\pi_\lambda(a|s)$ exist and induce a unique stationary distribution $\rho_\lambda(s, a)$. The primary objective in the average-reward setting is:

$$\theta^\pi \doteq \lim_{N \to \infty} \frac{1}{N} \mathbb{E}_{\tau \sim p, \pi, \mu} \left[ \sum_{t=0}^{N-1} r(\mathbf{s}_t, \mathbf{a}_t) - \alpha \log \pi(\mathbf{a}_t|\mathbf{s}_t) \right], \tag{1}$$

where $\alpha > 0$ is a temperature parameter controlling the strength of entropy regularization. Further details on average-reward RL can be found in Hisaki & Ono (2024); Adamczyk et al. (2025). This objective induces a Boltzmann distribution over trajectories: optimal policies assign higher probability to trajectories with larger cumulative reward Levine (2018). As a result, many high-level concepts from statistical mechanics such as free energy, temperature, and fluctuations admit direct analogues in MaxEnt RL. This formulation underlies modern algorithms such as Soft Q-Learning and Soft Actor-Critic Haarnoja et al. (2018a) and theoretical frameworks such as linearly solvable MDPs Todorov (2006) and its extensions Arriojas et al. (2023). In this work, MaxEnt RL serves two roles in connecting to the physical picture of non-equilibrium thermodynamics. First, it provides a clean probabilistic structure over trajectories that enables a closed-form analysis. Second, it allows the dynamical change in reward parameters to be interpreted as controlled deformations of an underlying distribution, making the interpretation of a curriculum as a "non-equilibrium driving protocol" precise.

## 3 A Thermodynamic Framing of Curriculum Learning

Suppose that reward functions are characterized by a finite-dimensional parameterization, $\lambda \in \mathbb{R}^L$: $r_\lambda(s, a)$. We assume task schedules, or curricula, $\lambda(t)$, represent (twice-differentiable) paths in the task space connecting two reward functions. The central question is then: *how should $\lambda(t)$ be chosen to minimize the total cost of adaptation?*

Below, we give a brief description of the framework used to address this question, with further details provided in Appendix A. To formalize the cost of adaptation, we track how the expected return achievable by the agent changes as the task parameters vary. Along a curriculum $\lambda(t)$, the total change admits an exact decomposition into a contribution from externally modifying the reward function and a contribution arising from the adaptation of the policy itself. Integrating this decomposition along the curriculum yields a path-dependent *excess work*, which vanishes only in the quasistatic limit. Interpreting the excess work as the cumulative cost of adaptation, we adopt its minimization as the objective for optimal curriculum design. We work in a quasistatic regime where task parameters vary slowly relative to the relaxation time of the policy-induced Markov chain, so that linear response theory applies. In this setting, we have the following approximation to the excess work Sivak & Crooks (2012):

$$\mathcal{W}_{\text{excess}} = \int_0^\infty \dot{\lambda}_i(t) \zeta_{ij}(\lambda(t)) \dot{\lambda}_j(t) \, dt, \tag{2}$$

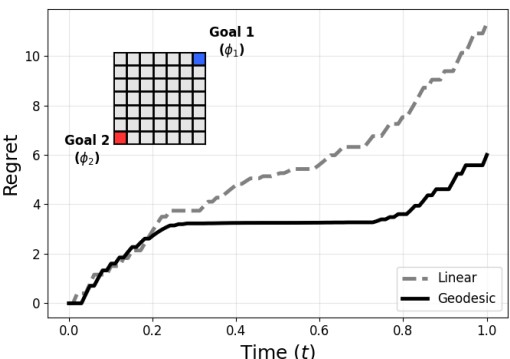 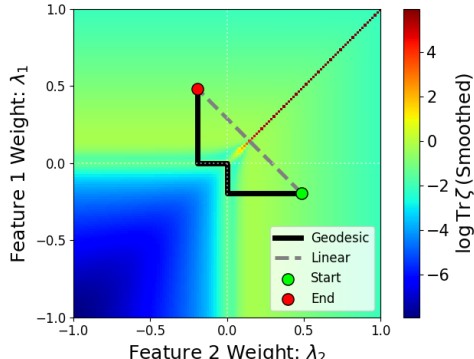

Figure 1: Visualization of the $7\times7$ Grid World task (inset, left), regret under the linear and geodesic paths (left), and the resulting thermodynamic manifold with the optimal geodesic protocol, which navigates around the phase transition at $\lambda_1 = \lambda_2$ (right). Further details can be found in Appendix B.

where we use Einstein summation notation, and $\zeta(\lambda)$ is a *friction tensor*: a symmetric, positive, semi-definite matrix given by the Green-Kubo relations,

$$\zeta_{ij}(\lambda) = \beta \sum_{t=0}^{\infty} \mathbb{E}_{\tau \sim p_\lambda}\left(\delta X_i(\mathbf{s}_t, \mathbf{a}_t) \cdot \delta X_j(\mathbf{s}_0, \mathbf{a}_0)\right), \qquad \delta X_i(s, a) = \frac{\partial r_\lambda(\mathbf{s}, \mathbf{a})}{\partial \lambda_i} - \mathbb{E}\left[\frac{\partial r_\lambda(\mathbf{s}, \mathbf{a})}{\partial \lambda_i}\right].$$

To move beyond heuristic reward parameter tuning or path sampling, we derive a method that allows controlled experimentation with task difficulty. That is, by defining this friction tensor, we convert the abstract notion of learning difficulty into a measurable geometric quantity. This allows us to predict where an agent will struggle, making the learning process more transparent. The entries of the friction tensor are defined through a two-point correlation function measuring how perturbations to the reward propagate under the Markov chain induced by the current (optimal) policy $\pi_\lambda$. Concretely, the quantities $X_i(s, a) \doteq \partial_{\lambda_i} r(s, a; \lambda)$ act as *generalized conjugate forces*, and $\zeta$ captures the relaxation time of the dynamics with respect to these forces. The quadratic form governing excess work endows the space of task parameters with a pseudo-Riemannian metric [1]. As a result, curriculum design reduces to a geometric optimization problem: choosing a path $\lambda(t)$ that minimizes length in task space. By standard variational arguments, via the Euler-Lagrange equations, optimal curricula must satisfy the geodesic equation:

$$\ddot{\lambda}^k + \Gamma_{ij}^k(\lambda)\, \dot{\lambda}^i \dot{\lambda}^j = 0, \tag{3}$$

where $\Gamma_{ij}^k$ is the Christoffel symbol associated with the metric. In general, these equations cannot be solved analytically, so we resort to numerical approaches and simplified settings to gain further insight into the resulting solutions. Solutions to Equation 3 yield optimal curricula which slow down in directions where the metric is large (costly adaptation) and accelerate where it is small, as evidenced by Figure 2 (inset) and Figure 1 (left). A notable consequence of this structure is that linear curricula are optimal only when the induced geometry is flat, i.e., $\zeta_{ij}(\lambda) = c$.

**Linear Reward Parameterizations.** An important and tractable example is when the reward function is linear in a set of (fixed) features $\phi(\mathbf{s}, \mathbf{a}) \in \mathbb{R}^L$, and thus $r(\mathbf{s}, \mathbf{a}) = \lambda^T \phi(\mathbf{s}, \mathbf{a})$. This "successor representation" has been discussed at length in the literature Dayan (1993); Barreto et al. (2017). The generalized forces are simply the features, $X_i = \phi_i(\mathbf{s}, \mathbf{a})$ and the metric simplifies to $\zeta_{ij}(\lambda) = \sum_{t=0}^{\infty} \mathbb{E}_{\pi_\lambda}\left[\widetilde{\phi}_i(\mathbf{s}_t, \mathbf{a}_t)\, \widetilde{\phi}_j(\mathbf{s}_0, \mathbf{a}_0)\right]$, where $\widetilde{\phi}_i = \phi_i - \mathbb{E}_{\pi_\lambda}[\phi_i]$ denotes the centered features. Crucially, the resulting geometry is generally curved (non-Euclidean) since the covariance depends on $\lambda$ through the policy $\pi_\lambda$. Consequently, the optimal curriculum is rarely a straight line; rather, it follows a geodesic that detours around regions of high feature variance to minimize thermodynamic cost. Figure 1 (right) illustrates an example in this regime. A linear path between two tasks (as specified by $\vec{\lambda}_0, \vec{\lambda}_1$) directly crosses the maximal friction $\lambda_1 = \lambda_2$.

---

[1]The matrix $\zeta$ is positive semi-definite, since there may be directions in task space (e.g. PBRS, linearly-dependent features) which do not affect the distribution $p_\lambda$.

## 4 TEMPERATURE ANNEALING

As an example, we apply the framework to temperature annealing in Maximum-Entropy RL (e.g., SAC) Haarnoja et al. (2018b), where the goal is to anneal the temperature $\alpha$ from a high value to a target $\alpha_T \approx 0$. Identifying the inverse temperature $\beta = \alpha^{-1}$ as the control parameter, we observe that $\beta$ scales the reward linearly, in which case the thermodynamic friction $\zeta(\beta)$, reduces to an auto-covariance of the rewards (see Appendix B for details). This quantity is computationally cheap and readily available during training.

Minimizing the excess work (the cumulative cost of adaptation) yields our proposed algorithm, "MEW", whose update rule follows $\dot{\alpha} \propto \alpha^2 / \sqrt{\sum \langle \delta r_k \delta r_{t+k} \rangle}$. This reproduces a classical result from finite-time thermodynamics: temperature should be changed slowly in regions of high variance and more rapidly when fluctuations are small Andresen & Gordon (1994). In the context of RL, this provides a principled mechanism for adaptive regularization. Rather than using a fixed decay, the entropy coefficient should dynamically "wait" (decay slowly) when the agent encounters high reward variability, and accelerate only when the policy's return stabilizes. We empirically examine this strategy in Figure 2. Here we find that the method scales to a high-dimensional locomotion task, Humanoid-v5 from Mujoco Todorov et al. (2012). In this example, the performance is stronger than the standard approach from (Haarnoja et al., 2018b), with more extensive experiments (e.g. on different environments) are left to future work. The standard protocol initially drops the temperature quickly, leading to a nearly-deterministic policy, which must be adjusted as the temperature later rises. On the other hand, our schedule is monotonic, and adjusts to the relative cost of adaptation at each step, allowing the policy to systematically adapt to fixed increments in friction. Our method also produces a protocol considerably more consistent between runs, as highlighted by the shaded region in Figure 2 (experimental details and further results are given in Appendix B due to space constraints).

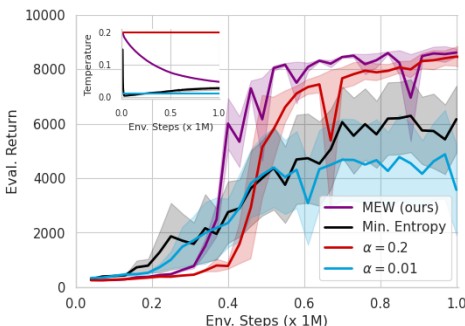

Figure 2: Proposed method, SAC's automatic temperature adjustment based on minimum entropy constraint, and two constant temperatures in Humanoid-v5 Towers et al. (2024). Base algorithm is ASAC Adamczyk et al. (2025).

## 5 DISCUSSION

In this work, we introduced a geometric framework for curriculum learning based on excess-work minimization, endowing task space with a pseudo-Riemannian structure that defines and guides optimal curricula. In doing so, we validated our hypothesis that optimal reward parameter schedules minimize the path-dependent excess cost from the friction tensor and follow geodesics in the induced task space. The resulting framework is readily applicable to the deep RL setting, as evidenced by the one-dimensional temperature annealing experiments shown in Figure 2 (see also Appendix B). Here, we found that the standard method for temperature reduction is considerably improved via the cooling schedule derived from our framework. More broadly, these results suggest that some empirical instabilities in RL may be understood not solely as algorithmic failures, but as consequences of driving a high-dimensional, non-equilibrium system too aggressively through a curved and evolving parameter manifold.

**Future Work** Several directions emerge from this work. On the theoretical side, further exploiting the induced geometry, and in particular clarifying the role of metric degeneracies, would extend the tools developed here. On the algorithmic side, developing scalable estimators of the friction tensor in deep reinforcement learning remains an important challenge. Finally, empirical validation on large-scale continual and lifelong learning benchmarks will be essential for assessing the predictive power of the proposed framework.

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

## A  THERMODYNAMIC CORRESPONDENCE

We briefly clarify the assumptions and setup required for our thermodynamic description of average-reward MaxEnt reinforcement learning.

1. **Average-reward MaxEnt formulation.** The optimal objective is the free-energy rate:
$$J(\pi_\lambda^*) = -\lim_{T \to \infty} \frac{F(\lambda)}{T} = \theta^{\pi_\lambda},$$
   where $\theta^{\pi_\lambda}$ is the entropy-regularized average-reward rate defined in Equation 1. Consider an average-reward maximum-entropy RL problem with a time-dependent reward
$$r_\lambda(s, a) = -H_\lambda(s, a), \qquad \lambda : [0, \mathcal{T}] \to \mathbb{R}^d,$$
   and inverse temperature $\beta$ (so that $\alpha = \beta^{-1}$). Let $\pi_\lambda^*$ denote the optimal policy at fixed $\lambda$, and let $p_\lambda$ denote the stationary trajectory distribution induced by $\pi_\lambda^*$.

2. **Ergodicity.** For each fixed $\lambda$, the Markov process induced by $\pi_\lambda^*$ admits a unique stationary trajectory distribution $p_\lambda$.

3. **Adiabatic tracking.** The agent's learned policy $\hat\pi_\lambda$ relaxes to $\pi_\lambda^*$ on a timescale $\tau_{\text{relax}}(\lambda)$, and the driving protocol satisfies
$$\|\dot\lambda(t)\| \ll \tau_{\text{relax}}(\lambda(t))^{-1}.$$
   Equivalently, on timescales over which $\lambda$ varies appreciably, the agent-induced trajectory distribution $p_t$ is close to the stationary distribution associated with the instantaneous task parameter $\lambda(t)$.

4. **Linear response.** Equilibrium time-correlation functions under $p_\lambda$ decay exponentially fast, and the protocol $\lambda(t)$ is twice continuously differentiable.

With these assumptions, we first derive the friction tensor, by mapping the reinforcement learning quantities onto those of the thermodynamic picture. Note that there are three timescales at play: (1) the MDP time, indexed by $k$, whose trajectories are used to calculate the friction; (2) the relaxation time (or solution time) to find an optimal policy at each stage in the curriculum; and (3) the curriculum time, denoted $t$. We write the latter with integrals to better distinguish its role from the other discrete time indices. We assume a separation of timescales: $\tau_{\text{MDP}} \ll \tau_{\text{relax}} \ll \tau_{\text{curriculum}}$. This can be ensured by tuning the curriculum speed to be sufficiently low relative to the RL learning rate, and waiting long enough for trajectories to relax for a given task when computing friction.

**Definition 1.** *The friction tensor is defined as:*
$$\zeta_{ij}(\lambda) = \beta \sum_{k=0}^{\infty} \mathbb{E}_{\tau \sim p_\lambda}[\delta X_i(s_k, a_k)\, \delta X_j(s_0, a_0)], \quad \delta X_i = \partial_{\lambda_i} r_\lambda - \mathbb{E}_{p_\lambda}[\partial_{\lambda_i} r_\lambda]. \tag{4}$$

In the following, we show how this definition follows naturally from the thermodynamic mapping. The definition for total work performed along the protocol is:
$$\mathcal{W} = \int_0^{\mathcal{T}} \dot\lambda(t)^\top \mathbb{E}_{p_t}[\partial_\lambda H_\lambda]\, dt = -\int_0^{\mathcal{T}} \dot\lambda(t)^\top \mathbb{E}_{p_t}[\partial_\lambda r_\lambda]\, dt.$$

The excess work is defined as $\mathcal{W}_{\text{ex}} = \mathcal{W} - \Delta F$, where $\Delta F$ is the change in equilibrium free energies (i.e. optimal entropy-regularized reward-rates): $\Delta F = F(\lambda(\mathcal{T})) - F(\lambda(0))$. This equilibrium quantity, which depends only on the endpoints of the path, represents the minimum possible cost, if the curriculum is driven infinitesimally slowly. We use the fundamental theorem of calculus and chain rule to rewrite $\Delta F = \int_0^{\mathcal{T}} F'(t) dt = \int_0^{\mathcal{T}} \left( \dot\lambda^T \cdot \nabla_\lambda \theta^{\pi_\lambda} \right) dt$.

By differentiating the trajectory-level partition function and invoking ergodicity, we obtain the thermodynamic identity: $\nabla_\lambda \theta^{\pi_\lambda} = \mathbb{E}_{p_\lambda}[\nabla_\lambda r_\lambda]$. When substituted into the excess work, we have:
$$\mathcal{W}_{\text{ex}} = \int_0^{\mathcal{T}} \dot\lambda(t)^\top \Big( \mathbb{E}_{p_\lambda}[\partial_\lambda r_\lambda] - \mathbb{E}_{p_t}[\partial_\lambda r_\lambda] \Big) dt. \tag{5}$$

Under the adiabatic tracking and exponential mixing assumptions, linear response theory allows the simplification of the instantaneous $(p_t)$ distribution in terms of the equilibrium distribution $(p_\lambda)$:

$$\mathbb{E}_{p_t}[\partial_\lambda r_\lambda] = \mathbb{E}_{p_\lambda}[\partial_\lambda r_\lambda] - \zeta(\lambda)\dot{\lambda} + \mathcal{O}\left(\|\dot{\lambda}\|^2\right), \tag{6}$$

where $\zeta$ is the integrated equilibrium autocovariance in Equation 4. Substituting Equation 6 into Equation 5 yields, to leading order,

$$\mathcal{W}_{\text{ex}} \approx \int_0^{\mathcal{T}} \dot{\lambda}(t)^\top \zeta(\lambda(t))\dot{\lambda}(t)\, dt. \tag{7}$$

This discussion follows the derivations in e.g. Sivak & Crooks (2012).

## B  EXPERIMENTS

In the continuous-control experiment (Fig. 2, Fig. 3) we use Humanoid-v5 as a prototypical high-dimensional RL environment. Since we operate in the average-reward regime, we use a variant of soft actor-critic designed for the average-reward setting: ASAC Adamczyk et al. (2025). As a baseline, we compare to (a) constant temperatures (low, $\alpha = 0.05$ and high $\alpha = 0.2$); (b) the dynamic temperature schedule introduced by Haarnoja et al. (2018b) (cf. Section 5 therein). Each run represents the mean over 5 independent runs, and the shaded regions represent one standard deviation. The friction is now a scalar function (there is one degree of freedom in specifying a task, $\alpha$). The friction is calculated with the following formula:

$$\zeta(\alpha) = \sum_{t=0}^{T} \mathbb{E}_{\pi_\alpha}[\delta r(s_t, a_t)\, \delta r(s_0, a_0)], \tag{8}$$

where we use the last $N(= 5000)$ transitions in the replay buffer. We note that this breaks the off-policy nature of soft actor-critic, since we use the fact that the replay buffer is storing samples from the learned policy. A more accurate, but more expensive procedure would involve rolling out the policy at every iteration, holding an on-policy replay buffer as well. The notation $\delta r$ denotes a centered reward: we subtract the mean reward (over all 5000 steps), as required by the definition of the friction. We use these approximations to ensure the algorithm works with minimal changes to the base algorithm. We use the authors' provided hyperparameters Adamczyk et al. (2025).

The empirical results in Figure 2 highlight that the traditional approach to temperature adjustment is not always stable, and sometimes over-aggressive in the induced temperature changes. Our new approach provides a principled method to decay temperature, while achieving strong performance. A more systematic experimental sweep over various environments and hyperparameters is left to future work.

In Figure 3, we show that our method is robust to the "thermodynamic speed" hyperparameter. From speeds of $10^{-7}$ to $10^{-4}$ we obtain a similar, strong performance. The speed implicitly adjusts the effective final temperature in the finite number of timesteps allowed. We also note that while our method monotonically reduces the temperature, there may be situations where re-heating the system is beneficial. The study of such cases and an algorithm for choosing dynamic target temperatures is the subject of future work. Pseudocode for the algorithm is shown below. Blue text denotes new changes required beyond the base algorithm.

### B.1  LINEAR FEATURES EXPERIMENT

To test the theory in a two-dimensional example, we set a linear reward function $r(s) = \lambda_1\phi_1(s) + \lambda_2\phi_2(s)$, where the features $\phi_i$ are one-hot encodings of opposing corners in a $7 \times 7$ grid (see Figure 1, inset). In Figure 1 (right), we calculate the friction tensor for $\lambda_i \in (-1, 1)$, using:

$$\zeta_{ij}(\lambda) = \sum_{t=0}^{T} \mathbb{E}_{\pi_\lambda}\left[\widetilde{\phi}_i(s_t, a_t)\, \widetilde{\phi}_j(s_0, a_0)\right], \tag{9}$$

with $T = 2000$ (steps for relaxation) and we average over the current policy's stationary distribution $(s_0, a_0) \sim \mu$ which is obtained by power iteration of the corresponding transition matrix, $P^{\pi_\lambda}$.

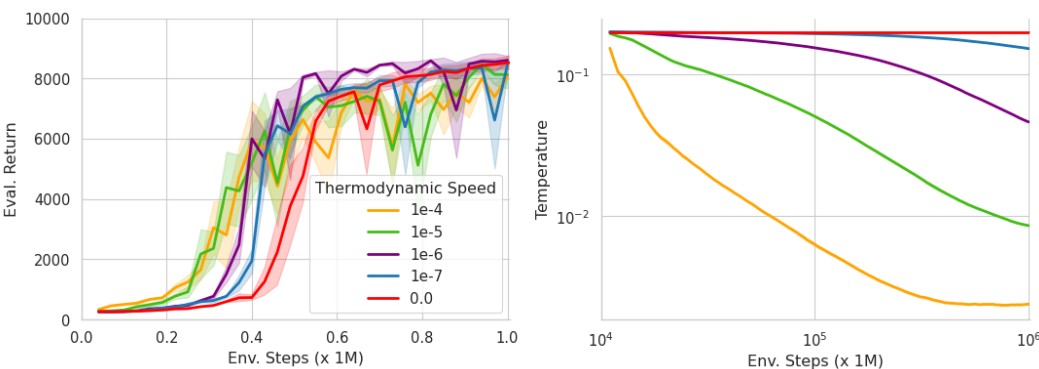

Figure 3: Comparison of different "thermodynamic speeds" (speed of traversal along the path from $\alpha = 0.2 \to 0$). Across three orders of magnitude, we obtain similar qualitative performance. The constant temperature, i.e. zero speed, $\alpha = 0.2$ is included for comparison. The log-log plot of temperatures indicates that a non-trivial protocol is learned. (If variance were constant throughout, a power law, i.e., a straight line, would result.)

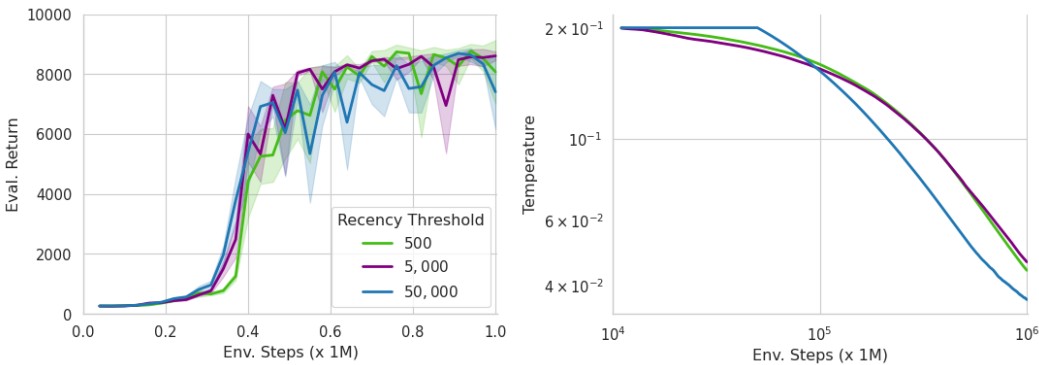

Figure 4: Comparison of performance for various "recency thresholds" (cf. $N$ in Algorithm 1). For an optimal policy, episodes are of length $1,000$; the recency thresholds used to calculate the friction therefore correspond to $1/2$, 5 or 50 episodes. The results shown here and in Figure 3 demonstrate that the proposed method is robust to the required hyperparameters.

We use tabular value iteration for the MaxEnt, average-reward RL objective. At each point in the parameter space, we reduce the $2 \times 2$ matrix to $\log \mathrm{Tr} \zeta$ (cf. Figure 3 of Rotskoff & Crooks (2015)) to plot a scalar representation of the friction. "Smoothed" indicates the application of a Gaussian filter with $\sigma = 0.1$. We restrict motion in the $\lambda$ plane to the cardinal directions along the chosen discretization (allowing diagonal motion yielded a qualitatively similar path). When $\lambda_1 = \lambda_2$ and $\lambda_i > 0$, the agent is attracted equally to either corner, resulting in a (soft) divergence of the friction tensor. This can be understood as the requirement for infinitely many samples to distinguish the goal of the agent. When either weight is negative ($\lambda_i < 0$) this does not occur: the agent avoids the corners. Temperature is set at $\alpha = 0.2$. Some results change in the low-temperature regime (i.e. divergence sharpens), but a further description of such phenomena are left to future work. On the left of Figure 1, we show the accumulated regret from either path, by calculating the optimal olicy at each stage of the protocol (discretized into 100 timesteps). The geodesic path incurs a lower regret.

## C  RELATED WORK

This work contributes to a growing literature that leverages geometrical and statistical structures to better understand learning dynamics. Prior work has explored the geometric formulations of policy optimization Kakade (2001), value functions Dadashi et al. (2019), and learning dynamics Lyle et al. (2022). Gleave et al. (2020); Wulfe et al. (2022) explore the definition of a metric in reward

---

**Algorithm 1** MEW: ASAC with Minimal Excess Work

---

**Require:** Thermodynamic speed $\eta$, recency threshold $N$, initial temperature $\alpha_0$
    Initialize actor $\pi_\phi$, critic $Q_\theta$, entropy-regularized reward-rate, $\rho$, replay buffer $\mathcal{D}$.
    Set inverse temperature $\beta \leftarrow 1/\alpha_0$.
    **while** not converged **do**
        Sample action $a_t \sim \pi_\phi(\cdot \mid s_t)$
        Observe $r_t, s_{t+1}$
        Store $(s_t, a_t, r_t, s_{t+1})$ in $\mathcal{D}$
        Sample batch $B \sim \mathcal{D}$
        Update critic parameters $\theta$
        Update reward-rate $\rho$
        Update actor parameters $\phi$
        Retrieve recent rewards $R_{\text{recent}} = \mathcal{D}[-K :]$   {Approximates on-policy data}
        Calculate $\zeta$ from Equation 8 using FFT {From Wiener–Khinchin theorem}
        $\Delta\beta \leftarrow \eta/(\beta\sqrt{\zeta + \epsilon})$
        $\beta \leftarrow \beta + \Delta\beta$
        $\alpha \leftarrow 1/\beta$
        Update target networks
    **end while**

---

space that accounts for the PBRS degrees of freedom (corresponding to zero friction in our setting); and (Huang et al., 2022) studied curriculum learning through the lens of optimal transport. This growing body of literature poses the curriculum learning problem as that of shifting one distribution to another, with least "work" (in the sense of the Earth mover's distance). Although our method shares this idea of shifting distributions (i.e. from one equilibrium distribution to another), the execution is distinct, as we consider a non-equilibrium driving protocol. Notably, we also do not assume access to the initial or final distributions in our experiments, and the protocol can, in principle, be calculated online (as done for the Humanoid experiments), by learning the friction tensor on the fly. Intuitively, the optimal transport results are based on how far apart tasks are; whereas our method is based on how difficult it is to move from one task to another. Our contribution therefore is to shift the focus to the parameterized task space, where we can leverage results from the physics literature to provide a principled framework for understanding how tasks can be varied during training.

