# OpenReview forum: "Thermodynamics of Reinforcement Learning Curricula"
_ICLR.cc/2026/Workshop/Sci4DL — Sci4DL 2026_

### Official Review · Reviewer_ak1P · 2026-02-18

**Fit:** 2
**Significance:** 2
**Confidence:** 2

**Summary:**

In a reinforcement learning context, identifying the optimal curriculum, or even defining a good curriculum, remains an open problem. This paper borrows intuition from non-equilibrium physics, arguing that changing a task forces the agent to adapt, which acts as "friction". Changing task parameters too fast in sensitive directions generates "excess work", representing wasted learning efficiency. Based on these concepts, the authors propose that an optimal curriculum minimises this excess work by following the shortest path through a defined task space, where the geometry is dictated by the friction tensor.

**Strengths:**

- The paper provides a principled, physics-inspired justification for scheduling task parameters and hyperparameters, offering a theoretical perspective on a challenging open problem.

- The theoretical framework is tested across two distinct settings: a 2D grid-world task that intuitively visualises geodesic paths across linear reward features, and a high-dimensional continuous control environment (Humanoid-v5), where the framework is applied to temperature annealing.

**Suggestions:**

- Clarity is a major issue in this paper. The paper is full of technical jargon that makes it hard to read even for people (like myself) with a general physics background. Overall, this compromises the paper's clarity and accessibility for a broader machine learning audience.

- The empirical validation is constrained to highly simplified low-dimensional parameterisations, ie a 2D linear reward space and a 1D inverse-temperature space. It is currently unclear how this framework can be generalised to complex multi-task learning environments, nor is it obvious how the appropriate continuous task space would be defined or computationally scaled in such settings.

---

### Official Review · Reviewer_DtD3 · 2026-02-27

**Fit:** 3
**Significance:** 2
**Confidence:** 2

**Summary:**

The paper proposes a geometric, thermodynamics-inspired framework for curriculum learning where a curriculum is viewed as a trajectory through a “task space” of reward (or task) parameters. The key idea is to define a pseudo-Riemannian metric on this space using a friction tensor derived from excess-work minimization, so that optimal curricula correspond to geodesics that minimize path-dependent excess cost.

The authors validate this perspective empirically in deep RL via one-dimensional temperature annealing, showing that their theoretically derived cooling schedule improves over a standard temperature-reduction heuristic. More broadly, they argue this lens suggests some RL instabilities may stem not only from algorithmic issues, but from driving a non-equilibrium learning system too aggressively through a curved, evolving parameter manifold.

**Strengths:**

Considering how central entropy scheduling is to good RL performance, the contribution of this work is interesting since it introduces an adaptive schedule that, being derived from a mechanistic model, provides a clear objective and interpretation for why this particular schedule should be preferred.

Moreover, the interpretation of curriculum learning provided by this work is particularly valuable because RL training dynamics are often hard to interpret and prone to instabilities. By framing instability as a consequence of driving a learning system too aggressively the paper offers a more diagnostic, interpretable lens that can help reason about when and why training breaks, and potentially guide more systematic interventions.

**Suggestions:**

Your derivation leverages a slow-driving/linear-response regime; have you considered extensions beyond linear response, for example protocols that include counterdiabatic-style corrections, and do you expect such corrections to matter in deep RL, where training can be highly non-stationary and parameter schedules may be relatively fast?

In practice, how robust do you think MEW would be to strongly non-stationary regimes? It would be interesting to see ablations showing when the system is slow enough for the friction estimate to be meaningful, and what failure modes appear when temperature is changed faster than what the policy and critic can relax.

Since MEW is compared to SAC’s automatic temperature tuning, it would help to clarify whether both methods were run under identical environment steps and update budgets, and to discuss how hyperparameter/tuning effort was balanced between them.

---

### Meta-Review · Area_Chair_ZJP2 · 2026-03-02

**Recommendation:** Accept

**Metareview:**

This paper defines a curriculum learning strategy for reinforcement learning based on non-equilibrium thermodynamics. An optimal curriculum corresponds to a trajectory in task space that minimizes the total cost of adaptation.
Although the empirical validation of the method is limited to two simplified settings, this physics-inspired viewpoint is interesting and a potentially valuable contribution to a challenging open problem.

---

### Decision · Program_Chairs · 2026-03-02

Accept